



# Response time correction of slow response sensor data by deconvolution of the growth-law equation

Knut Ola Dølven[1], Juha Vierinen[2], Roberto Grilli[3], Jack Triest[4], and Bénédicte Ferré[1]

[1]Centre for Arctic Gas Hydrate, Environment, and Climate,UiT The Arctic University of Norway, Tromsø, Norway
[2]Institute for Physics and Technology, UiT The Arctic University of Norway, Tromsø, Norway
[3]CNRS, University of Grenoble Alpes, IRD, Grenoble INP, 38000 Grenoble, France
[4]4H-JENA engineering GmbH Wischhofstrasse 1-3, 24148 Kiel, Germany

**Correspondence:** Knut Ola Dølven (knut.o.dolven@uit.no)

**Abstract.** Accurate, high resolution measurements are essential to improve our understanding of environmental processes. Several chemical sensors relying on membrane separation extraction techniques have slow response times due to a dependence on equilibrium partitioning across the membrane separating the measured medium (i.e., a measuring chamber) and the medium of interest (i.e., a solvent). We present a new technique for deconvolving slow sensor response signals using statistical

inverse theory; applying a weighted linear least squares estimator with the growth-law as measurement model. The solution is regularized using model sparsity, assuming changes in the measured quantity occurs with a certain time-step, which can be selected based on domain-specific knowledge or L-curve analysis. The advantage of this method is that it: 1) models error propagation, providing an explicit uncertainty estimate of the response time corrected signal, 2) enables evaluation of the solutions self consistency, and 3) only requires instrument accuracy, response time, and data as input parameters. Functionality

of the technique is demonstrated using simulated, laboratory, and field measurements. In the field experiment, the coefficient of determination ($R^2$) of a slow response methane sensor in comparison with an alternative, fast response sensor, significantly improved from 0.18 to 0.91 after signal deconvolution. This shows how the proposed method can open up a considerably wider set of applications for sensors and methods suffering from slow response times due to a reliance on the efficacy of diffusion processes.

**Keywords** Diffusion, Equilibrium, Error propagation, Membrane, Sensor, Laboratory experiment, Field experiment, Inverse methods

## 1   Introduction

High resolution *in situ* data are crucial to observe high variability in environmental processes when surrounding environmental parameters are continuously changing. Many contemporary measurement techniques have a limited response time due to signal





convolution inherited from a diffusion process, such as in Vaisala radiosondes (Miloshevich et al., 2004) or in continuous flow analysis of ice cores (Faïn et al., 2014). In oceanic sciences, measurement of dissolved analytes often requires an extraction technique based on membrane separation, where the property of interest (a solute) equilibrate across a membrane separating the medium of interest (a solvent) from the medium where the actual measurement takes place (a measurement chamber). This makes the sensor *response time* (RT) directly governed by how fast the analyte of interest can diffuse through the membrane.

This process is mainly driven by the difference in partial pressure between the two media and can be relatively slow. This results in high sensor RTs, leading to unwanted spatial and temporal ambiguities in recorded signals for sensors used in profiling (Miloshevich et al., 2004), on moving platforms (Bittig et al., 2014; Canning et al., 2021) or deployed in dynamic environments (Atamanchuk et al., 2015). Herein, we refer to sensors with this particular design as Equilibrium Based (EB) sensors and we seek to establish a robust, simple and predictable method for correcting high RT induced errors in data from

these sensors.

Considering an EB sensor during operation, we define $u_a(t)$ as the instantaneous ambient partial pressure of interest and $u_m(t)$ as the partial pressure within the measuring chamber of an EB sensor, where the measurement occurs. In this situation, a model of $u_m(t)$ as a function of time can be obtained via the growth-law equation (Miloshevich et al., 2004), which describes diffusion of the property of $u_a(t)$ through the separating barrier (in this case the membrane):

$$\partial_t u_m = k(u_a - u_m), \tag{1}$$

where $k$ is a sensor specific growth coefficient, which determines how fast a change in $u_a(t)$ will be reflected in $u_m(t)$. The RT of EB sensors are often given in $\tau_{63} = 1/k$, which corresponds to the time the sensor requires to achieve 63% (one e-folding) of an instantaneous step-change in ambient concentration. If $k$ in Eq. 1 is sufficiently small (i.e. $\tau_{63}$ is large), the diffusion will be slow and any fast fluctuations in $u_a(t)$, will be smeared out in time.

A numerical technique has already been proposed to recover fast fluctuations in $u_a(t)$ from measurements of $u_m(t)$ using a closed form piece-wise solution to Eq. 1 (Miloshevich et al., 2004). However, due to the ill-posed nature (see e.g. Tikhonov et al., 1977) of the forward model, errors in the measurements will be amplified when reconstructing $u_a(t)$. Miloshevich et al., (2004) counteracts this using an iterative algorithm that minimizes third derivatives to obtain locally smooth (noise-free) time-series prior to the reconstruction of $u_a(t)$. While this and similar methods seems to usually work well in practice (Bittig

et al., 2014; Canning et al., 2021; Fiedler et al., 2013; Miloshevich et al., 2004), it is difficult to determine the uncertainty of the estimate, as the iterative scheme does not model error behavior. Predicting the expected solution of the iterative estimator is also difficult and there is no straightforward way of choosing suitable smoothing parameters. These are important attributes for the reliability of solutions to these types of problems, due to the error amplification that occurs during deconvolution.

Herein, we establish an alternative method for estimating $u_a(t)$ from a measurement of $u_m(t)$. This solution is based on

the framework of statistical inverse problems and linear regression. Using a weighted linear least squares estimator, the growth law equation as measurement model, and a sparsity regularized solution, we are able to take into account uncertainties in the measurements, provide an intuitive and/or automated way of specifying an *a priori* assumption for the expected solution, and determine the uncertainty of the estimate. This approach also enables us to evaluate the self-consistency of the solution and





detect potential instrument and/or measurement issues. A time-dependent $k$ can also be employed, which suits membranes

with varying permeability (e.g. where $k$ is a function of temperature). We show that automated L-curve analysis produces well

regularized solutions, thereby reducing the number of input parameters to sensor response time, measurement uncertainty, and

the measurements themselves. The robustness/functionality of our technique was validated using simulated data, laboratory

experiment data, and comparison of simultaneous field data from a prototype fast response Diffusion Rate Based (DRB) sensor

(Grilli et al., 2018) and a conventional slow response EB sensor in a challenging Arctic environment.

## 60  2  Method

We assume that the relationship between observed quantity $u_a(t)$ and measured quantity $u_m(t)$ are governed by the growth-

law equation as given in Eq. 1. Estimating $u_a(t)$ from $u_m(t)$ is an inverse problem (Kaipio and Somersalo, 2006; Aster et al.,

2019) meaning that a small uncertainty in $u_m(t)$ will result in a much larger uncertainty in the estimate of $u_a(t)$, making it

impossible to obtain accurate estimates of $u_a(t)$ without prior assumptions.

To formulate the measurement equation (Eq. 1) as an inverse problem that can be solved numerically, we need to discretize

the theory, model the uncertainty of the measurements, and establish a means for regularizing the solution by assuming some

level of smoothness. We will denote estimates of $u_a(t)$ and $u_m(t)$ as $\hat{u}_a(t)$ and $\hat{u}_m(t)$. Measurements of $u_m(t)$ will be noted

as $m(t)$. Each element of the following steps is illustrated in Figure 1.

We discretize Eq. 1, using a time-symmetric numerical derivative operator:

$$70 \quad \frac{1}{2\Delta t}u_{i+1} - \frac{1}{2\Delta t}u_{i-1} + k_i u_i - k_i a_i = 0, \tag{2}$$

We have used the following short-hand to simplify notation: $u_m(t_i) = u_i$ and $u_a(t_i) = a_i$. Here $t_i$ is an evenly sampled grid of

times and $\Delta t_i = t_{i+1} - t_i$ is the sample spacing. We refer to $t_i$ as *model time* and for simplicity, assume that this is on a regular

grid $\Delta t_i = \Delta t$ with a constant time step. Note that the growth coefficient $k_i = k(t_i)$ can vary as a function of time.

We assume that sensor measurements $m_j$ of the quantity $u_m(t)$ obtained at times $t'_j$ (see Figure 1) have additive indepen-

dently distributed zero-mean Gaussian random noise:

$$m_j = u_m(t'_j) + \xi_j \tag{3}$$

where $\xi_j \sim \mathcal{N}(0, \sigma_j^2)$ with $\sigma_j^2$ the variance of each measurement, which in practical applications can be estimated directly from

the data or by using the known sensor accuracy. The $t'_j$ is the *measurement time*, which refers to the points in time where

measurements are obtained. We obtain $u_m(t_i)$ through gridded re-sampling of $m_j$ (see Figure 1). Note that the measurement

time-steps $t'_j$ do not need to be regularly spaced, nor coincide with the model times $t_i$.

To reliably estimate $u_a(t)$, we need to regularize the solution by assuming some kind of smoothness for this function. A

common *a priori* assumption in this situation is to assume small second derivatives of $u_a(t)$, corresponding to the second order

Tikhonov regularization scheme (Tikhonov and Arsenin, 1977). Although this provides acceptable solutions, the choice of the

regularization parameter (i.e. adjusting the amount of regularization applied) is not particularly intuitive. Since our method is





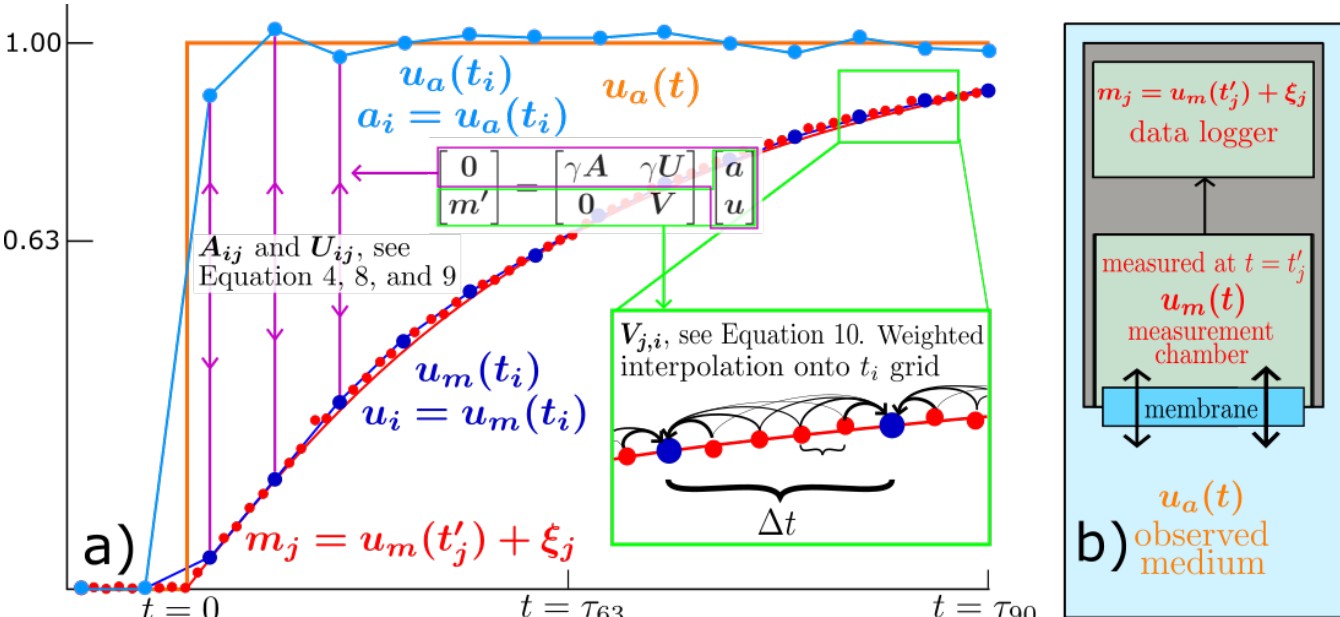

**Figure 1.** a) Schematic representation of Eq. 3-10 showing the relationship between the measurements ($m_j$, red dots) and de-convolution process for a step change in ambient property ($u_a$ orange line) lasting 2 e-folding times ($\tau_{63}$) of the diffusion equation. Thickness of arrows in zoomed inlet indicates weighting during re-sampling. b) Schematic representation of EB-sensor (grey box) during operation and physical location of the different properties in Eq. 1-10.

intended for a variety of domains where validation can be challenging, we have chosen to employ a different regularization method, where the regularization parameter relates directly to simple, real world characteristics and the ability of the instrument to resolve the ambient environment.

Model sparsity regularization (see e.g. Hastie et al., 2015) provides an intuitive model regularization by assuming that the observed quantity can be thoroughly explained by a reduced number of samples in some domain. In our case, we have used time domain sparsity, which translates to setting the number of model time ($t_i$) steps N smaller than the number of measurement time ($t'_j$) steps M. The *a priori* assumption we make to achieve this is that the observed quantity can only change with a time step of

$$\Delta t = \frac{\max(t_i) - \min(t_i)}{N - 1} = t_{i+1} - t_i, \tag{4}$$

and change piece-wise linearly between these points. Using this approach, an optimal regularization parameter becomes the lowest $\Delta t$ at which the observed quantity can change significantly. This means that choosing the regularization parameter can be done based on domain specific knowledge or scientific requirements for temporal resolution, within the limitations posed by the ill-posed nature of the problem and sensor performance.





We can now express the theory, relationship between the measurements and the theory, and the smoothness assumption in matrix form as follows:

$$\boldsymbol{m} = \boldsymbol{G}\boldsymbol{x} \tag{5}$$

$$\begin{bmatrix} \boldsymbol{0} \\ \boldsymbol{m}' \end{bmatrix} = \begin{bmatrix} \gamma\boldsymbol{A} & \gamma\boldsymbol{U} \\ \boldsymbol{0} & \boldsymbol{V} \end{bmatrix} \begin{bmatrix} \boldsymbol{a} \\ \boldsymbol{u} \end{bmatrix} \tag{6}$$

Here, $\boldsymbol{m}' = [\sigma_1^{-1}m_1, \sigma_2^{-1}m_2, \ldots, \sigma_M^{-1}m_M]^T$ contains the standard deviation normalized measurements and $\boldsymbol{a} = [a_1, a_2, \cdots, a_N]^T$ and $\boldsymbol{u} = [u_1, u_2, \cdots, u_N]^T$ the discretized concentrations $a_i = u_a(t_i)$ and $u_i = u_m(t_i)$ (see Eq. 2). $N$ is the number of model grid points $i$ and $M$ is the number of measurements $m_j$. $\boldsymbol{A} \in \mathbb{R}^{N \times N}$ and $\boldsymbol{U} \in \mathbb{R}^{N \times N}$ express the growth-law relationship be-
105 tween discretized $a_i$ and $u_i$ as given in Eq. 2. The matrix $\boldsymbol{V} \in \mathbb{R}^{M \times N}$ handles the regularization and the relationship between measurements and the discretized model of $u_m(t)$. The constant $\gamma \gg \sigma_j^{-1}$ is a numerically large weighting constant, which ensures that when solving this linear equation, the solution of the growth-law equation will have more weight than the measurements. In other words, the solution satisfies the growth-law equation nearly exactly, while the measurements are allowed to deviate from the model according to measurement uncertainty. The definitions of the matrices are as follows, where $k_i$ is the
110 growth coefficient:

$$\boldsymbol{A}_{ij} = \begin{cases} -k_i & \text{when} & i = j \\ 0 & \text{otherwise} \end{cases} \tag{7}$$

In matrix $\boldsymbol{U}$ we also need to express the time derivative of $u_m(t)$ and consider edge effects:

$$\boldsymbol{U}_{ij} = \begin{cases} k_i & \text{when} & i > 1 \text{ and } i < N \text{ and } i = j \\ (2\Delta t)^{-1} & \text{when} & i > 1 \text{ and } i < N \text{ and } j = i+1 \\ -(2\Delta t)^{-1} & \text{when} & i > 1 \text{ and } i < N \text{ and } j = i-1 \\ k_i - (\Delta t)^{-1} & \text{when} & i = 1 \text{ and } j = i \\ -(\Delta t)^{-1} & \text{when} & i = 1 \text{ and } j = i+1 \\ k_i + (\Delta t)^{-1} & \text{when} & i = N \text{ and } j = i \\ -(\Delta t)^{-1} & \text{when} & i = N \text{ and } j = i-1 \\ 0 & \text{otherwise} \end{cases} \tag{8}$$

The matrix $\boldsymbol{V} \in \mathbb{R}^{M \times N}$ relates concentration $u_i$ to measurements of this concentration $m_j$ (see Figure 1 and Eq. 2 and 3). We
use a weighted linear interpolation between grid points $t_i$ when assigning measurements to the model:

$$\boldsymbol{V}_{ji} = \begin{cases} (1 - |t_j' - t_i|/\Delta t)\sigma_j^{-1} & \text{when} & |t_j' - t_i| \leq \Delta t \\ 0 & \text{otherwise} \end{cases} \tag{9}$$

Where $\Delta t$ is the model timestep and regularization parameter (see Eq. 4). It is now possible to obtain a maximum *a posteriori* estimate of $u_a(t)$ and $u_m(t)$ by solving for the least-squares solution to matrix Eq. 5:

$$\hat{\boldsymbol{x}} = (\boldsymbol{G}^T\boldsymbol{G})^{-1}\boldsymbol{G}^T\boldsymbol{m}. \tag{10}$$





The vector $\hat{x} = [\hat{a}; \hat{u}]$ contains the maximum *a posteriori* estimate of vectors $a$ and $u$. The matrix $G$ is described in Eq. 5-9. The estimate $\hat{a}$ of $u_a(t)$ is the primary interest; however, the solution also produces an estimate $\hat{u}$ of $u_m(t)$. This can be a useful side-product for detecting outliers from fit residuals or other issues with the measurements (as shown in the field experiment).

     Equation 10 allows the estimate to also be negative, which can be unwanted if the observed quantity is known positive. In 125   such cases, it is possible to apply a non-negativity constraint using a non-negative least-squares solver (Lawson and Hanson, 1995).

     As the uncertainties of measurements are already included in the theory matrix $G$, the *a posteriori* uncertainty of the solution is contained in the covariance matrix $\Sigma_{\mathrm{MAP}}$, which can be obtained as follows:

$$\Sigma_{\mathrm{MAP}} = (G^T G)^{-1}. \tag{11}$$

This uncertainty includes the prior assumption of smoothness.

     The quality of the solution relies on an appropriate choice of regularization parameter $\Delta t$ and estimate of the noise/uncertainty in the measurements. We develop this through an application of the theory in a simulation experiment.

## 2.1   Simulation and $\Delta t$ determination

     To test the numerical validity of our method and develop a regularization parameter selection tool, we used a toy model. This 135   gives us the possibility to prove that the method gives well behaved consistent solutions as we know the correct results and control all input variables.

     We defined the simulated concentration $u_a(t)$ (see also Figure 1) as a step-wise change in partial pressure:

$$u_a(t) = \begin{cases} 0 & \text{when} \quad t < 5 \\ 1 & \text{when} \quad t \geq 5, \end{cases} \tag{12}$$

where units for time and partial pressure are arbitrary. While this is not a realistic scenario encountered under field conditions, 140   step-change simulations is a conventional calibration. It is also the most challenging scenario for testing our method, since it directly violates our smoothness assumption.

     The measurement chamber partial pressure $u_m(t)$ was simulated with a dense grid using a closed form solution of Eq. 1 from $u_a(t)$ using a growth coefficient $k = 0.1$ ($\tau_{63} = 10$). Sampling from $u_m(t)$ and adding Gaussian noise $\xi_j \sim \mathcal{N}(0, \sigma_j^2)$ gives the simulated measurements:

$$m_j = u_m(t'_j) + \xi_j \tag{13}$$

We assume that measurement errors are proportional to $u_m(t)$ in addition to a constant noise floor term, providing a standard deviation for each measurement given by:

$$\sigma_j = \epsilon u_m(t'_j) + \sigma_0 \tag{14}$$





We used $\epsilon = 0.01$ and $\sigma_0 = 0.001$ (0.1+1% uncertainty).

As $A$, $U$, and $m'$ of the matrix equation (Eq. 6) are now known, only the regularization parameter $\Delta t$ in $V$ (Eq. 9) needs to be defined to obtain the gridded estimate of time and RT-corrected measurements(i.e. $\hat{u}$ and $\hat{a}$) for the simulated model. For solutions regularized through smoothing, a well regularized solution captures a balance between smoothness and model fit residuals. Or in more practical terms, the provided solutions are sharp enough so critical information is not lost (i.e., detection of short term signal fluctuations are not removed by smoothing integration), but with reasonable noise and uncertainty estimates.

We have chosen a heuristic approach to this optimization problem, by applying the L-curve criterion (see Hansen, 2001) Statistical methods based on Bayesian probability (Ando, 2010), such as the Bayesian information criterion were also tested with similar results. We chose to apply the L-curve criterion due to its robustness and ability to intuitively display the effect the regularization parameter has on the solution, which we believe is an advantage in practical applications of our method.

     In our case, the L-curve criterion involves calculating a norm $E_s$, which measures how noisy the estimate is and a norm $E_m$,

which measures how large the fit residual is (i.e., an estimate of how well the model describes the measurements). These norms are calculated for a set of different regularization parameters, which are compared in a log-log plot (see Figure 2a) where the data points align to trace a curve that resembles the letter "L". The under-regularized (or too noisy) solutions are found in the upper left corner where perturbation errors dominate. The over-regularized (or over-smoothed) solutions are located in the lower right corner, where regularization errors dominate. Good regularization parameters are located in the middle of the bend

or kink of the L, where smoothness and sharpness are well balanced, limiting both noise and fit residuals.

     We have used the first-order differences of the maximum *a posteriori* solution $\hat{a}$ as the norm measuring solution noise:

$$E_s = \sum_{i=1}^{N-1} |\hat{a}_{i+1} - \hat{a}_i|^2 \tag{15}$$

and to approximate the fit residual norm, we use:

$$E_m = \sum_{j=1}^{M} |\hat{u}_m(t'_j) - m_j|^2 \tag{16}$$

where $m_j$ is the measurement of the quantity $u_m(t)$:

$$\hat{u}_m(t) = \sum_{i=1}^{N} w_i(t)\hat{u}_i \tag{17}$$

which comes directly from the least squares solution (Eq. 10) of matrix equation 6, where

$$w_i(t) = \begin{cases} (1 - |t - t_i|/\Delta t) & \text{when} \quad |t - t_i| \leq \Delta t \\ 0 & \text{otherwise} \end{cases} \tag{18}$$

corresponds to $V_{ji}$ (see Eq. 9) but without scaling for measurement error standard deviation. In essence, $\hat{u}_m(t'_j)$ is the best fit

model for the measured quantity $u_m(t)$ is obtained using linear interpolation in time from the least squares estimates in vector $\hat{u}$.





We calculated a set of estimates using a wide range of $\Delta t$ and produced the L-curve in Figure 2a using our norm and fit residual definitions (Eq. 15 and 16). We plotted estimates of $u_a(t_i)$ from $m_j$ using three different values of the $\Delta t$s shown in this L-curve to inspect an over-regularized, under-regularized, and well regularized solution. The error estimate is given as 95% confidence intervals. The solution in Figure 3a resulted from a low $\Delta t$=0.25 (upper left in the L-curve in Figure 2a) and is under-regularized and too noisy. The solution has small fit-residuals (Figure 2b), since the high resolution enables the model to represent almost instantaneous changes. Figure 3b, shows an over-regularized solution, with a high $\Delta t$=5 (lower right in the L-curve, Figure 2a). In this scenario, the noise is minimal; however, the coarse resolution gives poor fit with the pre-convolved signal. This can be clearly seen in (Figure 2c) as a spike in the fit residuals at the location of the step change ($t = 5$) where the model is unable to represent the process that produced our measurements due to the poor resolution. Choosing $\Delta t$ =1.35, located in the bending point of the L in Figure 2a, provides a well regularized solution, where sharpness is good enough to describe the step-change without too much noise or eye-catching spikes in the fit-residuals (Figure 2d).

Using $\Delta t$ =1.35, we can also inspect how a well-regularized solution is affected by edges and missing measurements (shown in Figure 3d). The missing measurements resulted in an increased uncertainty in the region where there are no measurements, but the maximum *a posteriori* estimate still provided a reasonable solution. Uncertainties grow near the edges of the measurement as expected, since there are no measurements before or after the edges which contain information about $u_a(t)$.

Choosing the best $\Delta t$ is a pragmatic task, where the L-curve criterion is a useful guideline. The kink in the L-curve can be chosen manually through visual inspection, or automatically by identifying the point of maximum curvature. We numerically approximated the maximum curvature location using a spline parameterization of the L-curve (method described in Appendix A) to find $\Delta t$ =1.35. Estimating $\Delta t$ using the L-curve criterion gives a solution with an numerically optimal compromise between noise and information about variability. However, increasing or decreasing $\Delta t$ slightly can be justified in instances where scientific hypotheses requires interpretation of very rapid variability given careful interpretation of the resulting solution. Nonetheless, if the regularization parameters $\Delta t$ that are found near the bend of the "L" are too large to meet the scientific requirements, this indicates that the measurement device is unable to resolve the phenomenon of interest due to low accuracy and/or too much convolution.

## 3  Laboratory experiment

We evaluated our proposed technique in a controlled laboratory experiment using a Contros HydroC $CH_4$ EB methane sensor by exposing the sensor to step changes (similar to the numerical experiment) in concentration. We connected the instrument to an air tight water tank (12.9 L) with a small headspace ($\sim$0.25L) via hoses where water was pumped at 6.25 L min$^{-1}$ and kept at constant temperature ($22^oC$) (see setup in Appendix B). The setup first ran for two hours to ensure stable temperature and flow. In this period, the sampling rate was 60 seconds, while it was changed to 2 seconds for the rest of the experiment to provide high measurement resolution for the step changes.

Four step-changes (two up and two down) were approximated by opening the lid and adding either 0.2 L of methane enriched or ultrapure water. The RT of the sensor was determined to 23 minutes at $22^oC$ (k=0.000725s$^{-1}$) prior to the experiment



**Figure 2.** a) L-curve for sweep of different $\Delta t$ values for estimated property (RT correction) of simulated measurements given by $m_j = u_m(t_j) + \xi_j$. The y-axis, $E_s$ is the noise in the data given by the step difference between adjacent data-points which is high for models that are too complex (lower $\Delta t$/higher number of data points) in the model. The x-axis is the fit error residual $E_m$, which shows how well the results (the $u_a$s) explains the measurements $m$. The latter is high for too sparse models (high $\Delta t$/low number of data points). b), c), and d) show fit residuals $\hat{u}(t_j') - m_j$ for each point in measurement time using $\Delta t$=0.25 (b), $\Delta t$=5 (c), and $\Delta t$=1.35 (d).



**Figure 3.** Estimated property using simulated data and different regularization parameters (time intervals). The simulated measurements $m_j = u_m(t_j) + \xi_j$ are shown with red points, and the estimated property $\hat{u}_a(t)$ is shown with a blue line with 2-$\sigma$ uncertainty indicated with a light blue line. Panel a) shows a high temporal resolution estimate, which is very noisy, due to the ill-posed nature of the deconvolution problem, b) shows a too low temporal resolution estimate, which fails to capture the sharp transition at $t = 5$, and c) shows a good balance between estimation errors and time resolution. Panel d) shows estimated property for a simulated data set with missing measurements between t = 15-17, which results in an increased error around the missing measurements for the estimated quantity. Error estimates are given as 95% confidence intervals.





following standard calibration procedure of the sensor, and parameters affecting membrane permeation was controlled (i.e. water flow rate over membrane surface and water temperature). We therefore used the signal noise calculated by the standard deviation of the single point finite differences as measurement uncertainty. Input uncertainty was lower for the first 2 hours as the lower sampling rate reduces signal noise due to a longer internal averaging period. $\Delta t$ was determined to 179 s using the automatic $\Delta t$ selection based on the L-curve criterion (see Appendix A). This also gave well confined fit residuals, with

a difference in signal noise at around 7000 s due to the change in input noise (caused by changes in sampling rate from 60 seconds to 2 seconds after 2 hours, Figure 4a and b).

At the first step change, the RT-corrected concentration rapidly increased from around 2.6 $\mu$Atm to $\sim$41 $\mu$Atm (Figure 4c and d). This was followed by a slower increase taking place over around 30 minutes up to $\sim$47 $\mu$Atm and then a slow decrease for another 30 minutes down to $\sim$45 before the next step change. The following step increase and subsequent two

step decreases followed the same pattern, which can be explained by three processes indicated in Figure 4d): The first rapid increase (process 1) results from the initial turbulent mixing caused by the abrupt addition of methane enriched water to the tank. This is followed by a slower diffusive mixing phase occurring after the water has settled (process 2). While these processes are occurring, there is also a gradual diffusion of methane to the headspace, which is shown as decreasing concentrations in approximately the last half of the plateau periods (process 3). As expected, this decrease was faster for higher concentrations,

due to the larger concentration gradients across the water/headspace interface. The step decreases show the same behavior, although with process 2 inverted.

Estimated uncertainty of the RT-corrected data averaged 0.64 $\mu$Atm (95% confidence) which is roughly double the raw data noise of 0.29 $\mu$Atm. Due to the long concentration plateaus, the balanced $\Delta t$ lies in a quite strongly regularized solution. Nonetheless, the de-convolved instrument data gives a considerably better representation of the step-changes with a relatively

small uncertainty estimate and reveals known features of the experiment setup (processes 1-3 in Figure 4d) which is obscured in the convolved data.

## 4  Field experiment

Continuted evaluation of our proposed technique was applied under more challenging conditions in a field based study using simultaneous data from two different methane sensors towed over an intense seabed methane seep site offshore West Spitsber-

gen (Jansson et al., 2019). A slow response EB Contros HISEM CH$_4$ and a fast response Membrane Inlet Laser Spectrometer (MILS) DRB sensor (Grilli et al., 2018) were mounted on a metal frame and dragged at various heights over the seabed ($\sim$20-300 m) at 0.4-1.1 m s$^{-1}$ in an area with many hydro-acoustically mapped methane seeps. The rapid and large variability in methane concentration, direct comparison with the DRB sensor, and the particularly high RT of the EB sensor in cold water made this an ideal test scenario for field based applications.



**Figure 4.** a) L-curve for sweep of different $\Delta t$ values for estimated RT corrected EB sensor data. The y-axis, $E_s$, is the noise in the data given by the step difference between adjacent data-points which is high for models that are too complex (lower $\Delta t$/higher number of data points) in the model. The x-axis is the fit error residual, $\hat{u}_m(t'_j) - m_j$, which shows how well the best fit model explains the measurements. b) Fit residuals $\hat{u}_m(t'_j) - m_j$ (black line) for each point in measurement time using $\Delta t=179$ s. c) show the result of the deconvolution (black line), uncertainty estimate (grey shading) and raw EB sensor data (blue line). d) is a zoomed in version of c) with area specified in c).





### 4.1 Growth coefficient and measurement uncertainty

We determined the growth coefficient $k$ (or the inverse, the $\tau_{63}$) for the EB sensor prior to the field experiment (see Appendix C) to be $5.747e^{-4}$ s$^{-1}$ ($\tau_{63}$=1740 s) at 25$^o$C. Taking the temperature dependency for the permeability of the polydimethylsiloxane sensor membranes into account (Robb, 1968) we found the following relationship for $k$:

$$k(T) = k_0 + \alpha_k T \tag{19}$$

where $k_0$ =$3.905e^{-4}$ s$^{-1}$ is $k$ at temperature T= 0$^o$C and $\alpha_k$ =$7.38e^{-6}$s$^{-1}$ $^o$C$^{-1}$ ($4.200e^{-4}$s$^{-1} \leq k \leq 4.377e^{-4}$s$^{-1}$ for water temperature 4$^o$C$\leq$T$\leq$6.4 $^o$C in the field experiment). We did not take the RT of the DRB sensor into account in the comparison, since its $\tau_{63}$ was negligible (8.0 s$< \tau_{63} <$8.3 s at 25$^o$C) compared to the EB sensor.

The measurement uncertainty ($\xi_j$ in Eq. 14) was set to either the estimated raw data noise or to the stated sensor accuracy after equilibrium is achieved, depending on which of these parameters was higher. The EB accuracy is stated to 3% using the ISO 5725-1 definition of accuracy, which involves both random and systematic errors. High concentrations during the field experiment made the 3% sensor accuracy our main input parameter for measurement uncertainty.

### 4.2 $\Delta t$ determination

We produced an L-curve from a set of estimates of $u_a(t)$ with $\Delta t$s ranging from 10 to 550 seconds (Figure 5a). Using a polynomial spline to approximate the maximum curvature point (see Appendix A) we found the optimal $\Delta t$ to be 55 s, which is in good agreement with our visual inspection of the L-curve. Upon inspection of the fit residual plot (Figure 5b), we observed large spikes at several time points, meaning that the model failed to describe the transformation between the measurements $m_j$ and the RT-corrected estimate $u_a(t_i)$. Inspection of raw data (red line in Figure 5b) uncovered sharp signatures in the measured dissolved concentration at these instances - too sharp to be a real signal (as these should have been convolved by the instrumental function). We concluded that there was an unidentified problem with the EB sensor system at these instances, most likely related to power draw, pump failure or other instrumental artifacts. Removing these problematic sections and redoing the estimates provided an L-curve with an unchanged maximum curvature location. We therefore kept using $\Delta t$=55 s in the final deconvolution which now gave a solution with approximately Gaussian distributed fit residuals without spikes (Figure 5d), meaning that we have a self-consistent and valid solution.

### 4.3 Sensor data comparison

The comparison between DRB, EB, and RT-corrected EB sensor data collected during the transect offshore west Spitsbergen is shown in Figure 6. The untreated EB sensor data clearly show how the convolution creates a strong hysteresis effect and makes the sensor unable to directly detect rapid changes in methane concentration. This results in a low coefficient of determination ($R^2 = 0.18$), high Mean Absolute Error (MAE = 9.77 ppm), and flat slope angle ($\alpha = 0.47$) when compared to DRB data (Figure 6a and b). RT-corrected EB data, on the other hand, match well with the DRB data ($R^2$=0.91, MAE = 4.1 and $\alpha$=0.82, Figure





**Figure 5.** a) L-curve for sweep of different $\Delta t$ values for estimated RT corrected EB sensor data. The y-axis, $E_s$, is the noise in the data given by the step difference between adjacent data-points which is high for models that are too complex (lower $\Delta t$/higher number of data points) in the model. The x-axis is the fit error residual, $\hat{u}_m(t_j') - m_j$, which shows how well the best fit model explains the measurements. b) Fit residuals $\hat{u}_m(t_j') - m_j$ (blue line) for each point in measurement time using $\Delta t$=55 s and raw EB sensor data (red line). Residual spikes attributed to a malfunctioning of the EB sensor are indicated by the black arrows (and zoom in oval inlet) c) and d) reported the results from the same analysis as a and b, but on the dataset where the problematic regions defined in figure b were removed.





6a and c), showing that high resolution data was indeed convolved in the EB data and that our method managed to retrieve them successfully.

The high $R^2$ of the RT-corrected data confirms that the EB sensor captured most of the variability in dissolved methane; however, there is a slight bias in the differences between the two data sets ($\alpha$=0.82). Inspecting the absolute differences reveals that the RT-corrected EB data have more moderate concentrations during periods of very strong variability. This can partly be explained by the inherent smoothing of a sparse model (for larger $\Delta t$). In theory, increasing the model complexity (reducing $\Delta t$) should return a slope closer to 1 and reduce differences. However, our attempt at decreasing $\Delta t$ for achieving this did not improve the slope, but increased the noise as expected. The flat slope could also at least partly originate from the previously problematic sections (spikes in Figure 5) in the EB sensor data (arrows in Figure 5b). Even though we ignored the data at these intervals, the offset in absolute concentration still affects our end-estimate. Another explanation could be an overestimation of the $k$ of the EB sensor in the laboratory procedure prior to the field campaign. Indeed, when using a lower $k$, e.g. corresponding to $\tau_{63}$ =3600 s (keeping $\Delta t$=55 s), which matches better with calibration results for similar sensors, the slope goes to 1 and differences are evenly distributed between high and low methane concentrations. The small slope offset could also be a combined effect of the above iterated reasons.

Using the framework of inverse theory allows us to model error behavior, enabling a comparison of the uncertainty estimates of the two sensors (shaded regions in Figure 6). For the DRB sensor data, we used the stated 12% accuracy as the uncertainty estimate (Grilli et al., 2018). For the RT-corrected EB data, we used the 95% confidence of the deconvoluted estimate using input measurement uncertainty, resulting in a median uncertainty range of 22%. Linearly interpolating the RT-corrected EB data onto DRB data time and taking the mutual error bounds into account, the two data sets agree within the uncertainties 92% of the time. This is despite the lower resolution of the RT-corrected data and other error sources (some of them described above) and we consider this a successful result.

The DRB data has a lower median relative (%) uncertainty estimate, but to compare these relative uncertainty estimates directly can be slightly misleading as the relative uncertainty estimate of the RT-corrected data varies in time (Figure 6d). This is due to the EB sensor convolution occurs *prior* to the time when the actual measurement (including measurement uncertainty) takes place (see Figure 1b). Since input measurement uncertainty mostly follows 3% of measured (already convolved) value, the uncertainty estimate becomes a function of the EB raw data. Consequently, due to the raw data hysteresis, the uncertainty becomes lower for increasing concentrations compared to decreasing concentrations, and vice versa. We can observe this at ∼00:15 and ∼07:15, when the RT-corrected concentration increases dramatically, but the uncertainty estimate is still relatively small (Figure 6a and d). On the other hand, between 04:40 and 05:30, the RT-corrected concentration data is relatively low and constant, but the uncertainty estimate is large and shrinks slowly due to the slow decrease in $u_m(t)$. Comparing the error bounds of the data-sets using the relative uncertainty is therefore a simplification because of the raw-data -inherited RT-corrected uncertainty estimate. Overall, this adversely affects the relative uncertainty estimate of the RT-corrected data set, since high error bounds inherited from the raw-data hysteresis during decreasing concentrations is divided by RT-corrected low concentration values which results in high relative uncertainties. A more narrowly defined (if possible) input uncertainty estimate for the EB sensor could help constrain this uncertainty.





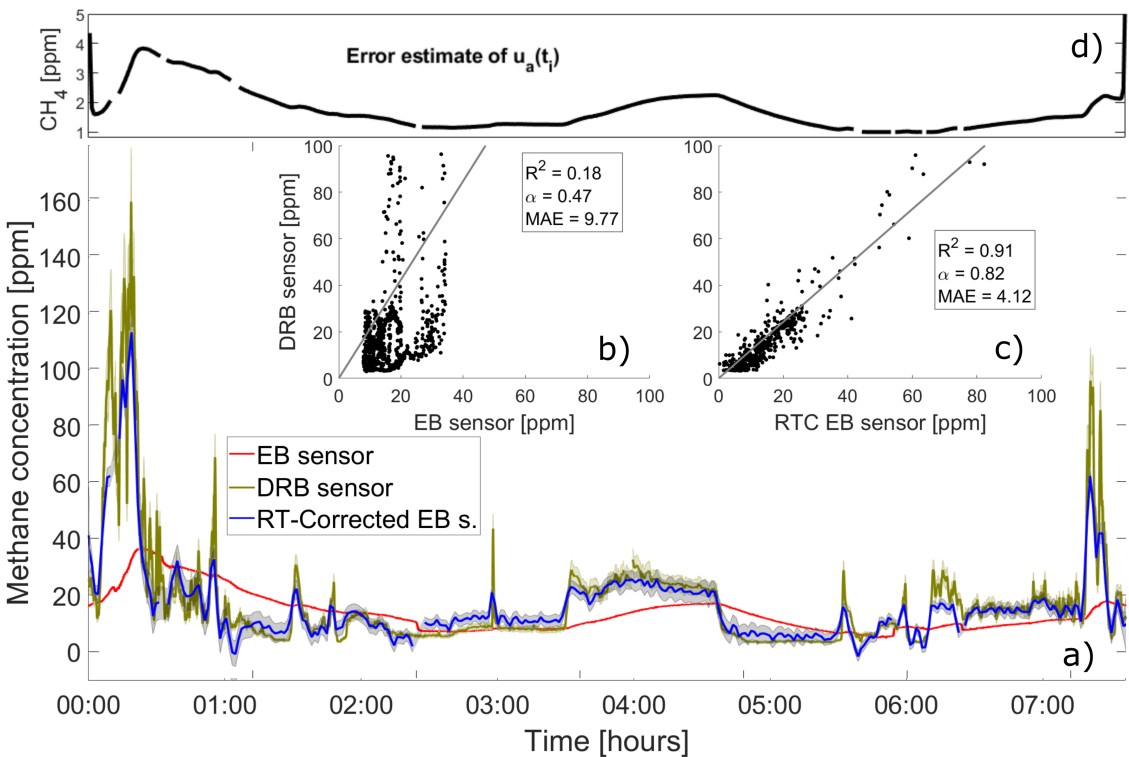

**Figure 6.** a) Field data from the DRB (yellow), EB (red), and RT-Corrected EB (blue) sensors. b) and c) show direct comparison between DRB and the EB sensor data. Coefficient of determination ($R^2$), Mean Absolute Error (MAE), and slope angle ($\alpha$) is given for comparison between the DRB sensor data and either raw (b) or RT-corrected (c) EB sensor data. d) show the error estimate of the RT-corrected signal, $u_a(t_i)$.

## 5 Conclusions

We presented and successfully applied a new RT-correction algorithm for membrane based sensors through a deconvolution of the growth-law equation using the framework of statistical inverse problems. The method requires few and well-defined input parameters, allows the user to identify measurement issues, models error propagation and uses a regularization parameter which relates directly to the resolution of the response time corrected data. Functionality testing was done using both a laboratory and a field experiment. Results from the laboratory experiment uncovered features of the experimental setup which were obscured by convolution in the raw data and the field experiment demonstrated the robustness of the algorithm under challenging environmental conditions. In both tests, the sensors ability to describe rapid variability was significantly improved and better constraints on input uncertainty and response time are areas which can potentially further enhance results.

This method and validation experiments using the Contros/HISEM sensors uncovers a new set of applications for these and similar sensors, such as ship-based profiling/towing and monitoring highly dynamic domains. Conventional EB sensors are also more abundant and affordable compared to more specialized equipment, increasing the availability and possibilities for





scientists requiring high resolution data to solve their research questions. Additionally, we believe this deconvolution method could be applicable to other measurement techniques as well, where diffusion processes hampers response time.





## Appendix A: Automatic $\Delta t$ selection

Even though $\Delta t$ sometimes can be chosen purely based on the practical problem at hand, we also want to provide a more rigorous way of choosing $\Delta t$ applicable at any circumstance. There are several ways to approach this problem (see e.g. Ando, 2010), but we have used the L-curve criterion. Even though $\Delta t$ can be chosen through visual inspection of the L-curve, we also provide the option of automatic $\Delta t$ selection to further simplify and provide more robustness to the methodology. This is done by finding the point of maximum curvature in the L-curve, which corresponds to the kink of the L. We do this by fitting a 4th degree smoothness regularized cubic spline to a sweep of a given number of solutions estimated using evenly distributed $\Delta t$s between a $\Delta t$ corresponding to one half of the measurement time-step up to a maximum of 2000 model points and a $\Delta t$ corresponding to a 10 point model grid. A $\Delta t$ located in the bend of the L can then be found by using the derivatives of the polynomials in the spline and maximizing the curvature given by

$$K = \frac{S'_{E_m} S''_{E_s} - S''_{E_m} S'_{E_s}}{(S'^2_{E_m} + S'^2_{E_s})^{\frac{3}{2}}},$$

where $S_{E_m}$ and $S_{E_s}$ are the splines of $E_m$ and $E_s$ and using Lagrangian differential notation. Smoothing is done by including the second derivatives weighted by a smoothing parameter in the minimization criteria of the spline fit.

One issue that arose during development of the automatic $\Delta t$ selection algorithm was that it was applied to data from a toy model where we tried to estimate a step-change in property (Figure 3 in manuscript). This is of course an unrealistic situation to encounter in any field application of a real instrument and is also in violation of any smoothness assumption we make on the solution. More specifically, we assume that changes in $u_a(t)$ can only occur following a piece-wise linear model with a time resolution of $\Delta t$, which is violated in the case of an instantaneous step-change. The L-curve criterion will nonetheless give us the best possible approximation we can get to the most likely solution of our problem. However, the fit residuals between $\hat{m}_j$ and $m_j$ will be dependent on the match or mismatch between the time-steps in $u_a(t_i)$ (with resolution defined by $\Delta t$) and the time when the instantaneous step change occurs. In essence, if there is a good match between the model time-steps and the instantaneous step-change, the model will be able to produce lower fit-residual and vice-versa. The result of this is that the spline fit can, depending on the location of the knots, produce local points with very high curvature which are not located in the kink of the L. We counteracted this effect when using the toy model data by doing a simple running mean and sorting of the noise and fit residual data for the solutions produced during the $\Delta t$ sweep, which resulted in consistent results. In a real world application where there is constant, but less abrupt variability, this should not be an issue, but we kept the running mean filter to increase the robustness of this approach. We also compared the automated model selection based on the L-curve criterion as iterated herein with model selection based on the Bayesian information criterion (see e.g. Ando, 2010), which gave similar results. Nonetheless, it is recommended that the ability to visually inspect the L-curve is exploited, to ensure the automatic selection has worked as expected.





**Appendix B: Laboratory setup**

**Figure B1.** a) Schematic representation of the experiment setup and b) picture of the experiment setup. The tank had a air tight dome shaped lid with a small headspace (∼0.3 L) and was 27.5 cm high and 24.5 cm diameter. Room temperature was controlled and kept constant.





## Appendix C: Growth coefficient determination for field experiment

To apply our methodology to the EB field sensor data set and compare with the DRB data, the sensor growth coefficients $k$ (or $\tau_{63}$) are required. Since no calibration data was available on location, we estimated $k$ directly prior to the field campaign by placing both instruments in a freshwater filled container (25 L, 25°C), where $\sim$500 ml of methane enriched water was added instantaneously to simulate a step change in concentration. The water was continuously mixed using the two submersible pumps

provided by the instruments and corrected for degassing to the atmosphere. With this setup, $k$ was estimated to $5.747e^{-4}$ s$^{-1}$ ($\tau_{63}$=1740 s) for the EB sensor and $7.69e^{-2}$s$^{-1}$ ($\tau_{63}$=13 s) for the DRB sensor.

Water temperature and salinity have a direct impact on $k$ for both these sensors due to changes in gas permeation of the membrane Robb (1968). This makes $k$ a function of time in a field experiment where these properties are varying. Based on laboratory testing on the permeation efficiency of the polydimethylsiloxane (PDMS) membranes used in both sensors (see

Grilli et al., 2018) we found that $k$ increased linearly with temperature following

$$k(T) = k_0 + \alpha_k T$$

where $k_0$ is $k$ at T= $0^o$C and $\alpha_k$ is a constant individually determined for each sensor. The effect of salinity was negligible at the low water temperatures at our field study site. To keep the RT of the fast response sensor as low as possible during the field campaign, we increased the total gas flow in the DRB sensor, thereby counteracting some of the loss in responsiveness

due to lower water temperature. The water temperature range during the field study was 4.0-6.4$^o$C, which gave a $k$ between $4.200e^{-4}$ and $4.377e^{-4}$s$^{-1}$ ($\tau_{63}$ =2285-2381 s) for the EB sensor and between 0.120 and 0.125 s$^{-1}$ ($\tau_{63}$ =8.0-8.3 s) for the DRB sensor, taking increased gas flow into account.


*Code and data availability.* All data and code presented in this paper can be obtained upon request to the authors and will be made available in the platform Open research Data at the University of Tromsø – The Arctic University of Norway (https://dataverse.no/dataverse/uit) and

will accompany the manuscript upon final publication.

*Author contributions.* Conceptualization: KOD,JV,RG,JT,BF. Data curation: KOD,RG. Formal Analysis: KOD,JV,RG. Funding acquisition: BF,RG,JT. Investigation: KOD,RG,JT. Methodology: KOD,JV. Project administration: KOD,RG,JT,BF. Resources: RG,JT,BF. Software: KOD,JV. Supervision: KOD,RG,JT,BF. Validation: n/a. Visualization: KOD. Writing - original draft preparation: KOD,JV. Writing - review & editing: KOD,JV,RG,JT,BF.

*Competing interests.* The authors declare that they have no conflict of interest.

*Acknowledgement.* The research leading to these results has received funding from the European Commission's Seventh Framework Programmes ERC-2011-AdG under grant agreement no. 291062 (ERC ICE&LASERS), the ERC-2015-PoC under grant agreement no. 713619 (ERC OCEAN-IDs) and the Agence National de Recherche (ANR SWIS) under grant agreement no. ANR-18-CE04-0003-01. Additional funding support was provided by SATT Linksium of Grenoble, France (maturation project SubOcean CM2015/07/18). This study is a part

of CAGE (Centre for Arctic Gas Hydrate, Environment and Climate), Norwegian Research Council grant no. 223259). We thank the crew of R/V Helmer Hanssen, Pär Jansson for initial discussions of manuscript idea, Snorre Haugstulen Olsen for input on the automatic $\Delta t$ selection, and Nick Warner for final proofreading and general feedback on the content of the manuscript.





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
