# Peer review of "Response time correction of slow response sensor data by deconvolution of the growth-law equation"

_Geoscientific Instrumentation, Methods and Data Systems, 2021_

## Referee Comment (RC1)

The paper by Dølven et al. entitled "Response time correction of slow response sensor data by deconvolution of the growth-law equation" deals with the reconstruction of an ambient signal from a slow response/low pass filtered sensor time series. They present the theoretical framework of their approach and apply it to three examples, a step change idealized toy model as well as a laboratory and a field experiment. While the examples center around a slow response (equilibration-based) methane sensor, the scope of the respone time correction method goes beyond those specific applications and can be applied to a wide range of similar scenarios (as the authors note in the introduction).

What makes this work unique is that it (1) derives information on the level of smoothing that should be applied to the reconstruction of the deconvolved signal from the data themselves, and (2) provides an uncertainty bound to the reconstruction, all while having only minimal requirements of number of input parameters/assumptions on sensor behaviour. This sets it apart from previous work (e.g., Miloshevich et al., 2004) with similar goals, and the current work promises to have a large impact on the field going community.

The work is logically structured and written concisely and to the point, sometimes a bit brief considering the likely non-math/signal processing audience of the article. Nonetheless, it is one of the most excellent works I reviewed recently.

**Comments**

1. In your simulated data and toy model, you use a measurement error proportional to $u_m$ plus a constant noise floor term. After reading the full paper, I see where this comes from (your particular application/example), but do not think that it serves your objectives here as outlined in section 2.1 (l. 134ff.). While the lagged inflation/deflation of the estimate's $\hat{u}_a$ uncertainty is well explained and understandable in the field example (Figure 6d and corresponding text), it raises question marks and creates wrong conclusions here, early on in your derivation, e.g., "*So it seems that the method accumulates/amplifies noise terms with time, as the $\hat{u}_a$ uncertainty envelope grows and grows with time (Figure 3a and c). What would happen if there were a longer 0-period of say time 50 instead of time 5, would the error envelope be huge already at the step change? Can the method only be applied to short pieces of a longer time series??*"

    Instead, I'd suggest to use only a constant measurement error term in the toy model for simplicity, to illustrate that your method does not suffer from accumulation of noise and (in such a case) can keep a nearly constant $\hat{u}_a$ uncertainty. *However*, ...

2. I find it hard to understand and believe that (e.g., with a constant $\sigma_j$) the uncertainty of the reconstruction $\hat{u}_a$ does not have/show a dependence on the magnitude of correction (e.g., difference between $\hat{u}_a$ and $\hat{u}_m$). I.e., that the reconstructed signal of a sensor pretty much in equilibrium (in

no-gradient regions) should be as certain or uncertain as the reconstructed signal where the sensor experiences a strong gradient (and needs a strong correction).

Intuitively, with small corrections required (no-gradient case), sensor noise should not be of paramount importance (just slight amplification, reflected in smaller estimate's uncertainty envelope), whereas with large corrections (strong gradient case), the same level of sensor noise should have a stronger impact on the reconstructed signal (more strongly amplified, so larger estimate's uncertainty envelope), shouldn't it?

The figures 2 and 3 tend to support this intuition: "Overshoots" in the reconstructed signal $\hat{u}_a$ tend to be a more prominent issue in strong gradients (here: step change, which violates the smoothness assumption as noted in the Appendix A, agreed. But the same is true if $u_a$ has a "smooth" strong gradient within $\Delta t_j$ resolution). Similarly, fit residuals (Figure 2d) show a different character at the step change/strong gradient, where they are more coherent (and more coherently wrong) across multiple $t_j$ (suggesting $\hat{u}_a(t_i)$ uncertainty should be larger), whereas they show only high frequency fluctuations around a zero-mean in the later no-/small gradient (suggesting validity of a smaller $\hat{u}_a(t_i)$ uncertainty).

This is probably the most important point that needs to be addressed by the authors (and testifies to the otherwise really excellent work): Should the uncertainty on $\hat{u}_a$ not be larger when there is a large correction compared to a small one?

3. The authors present examples where there are highly-resolved observations together with a very long time constant. This is arguably a very favourable case for response time corrections, where the high resolution allows for a lot of averaging, thus keeping noise amplification low.

   The authors should add a comment in their manuscript on more poorly-resolved scenarios, where $\tau_{63}$ is closer to $\Delta t_j$ and not orders of magnitude between them ($\tau_{63} \gg \Delta t_j$). Is there a limit (e.g., to the applicability, utility, ...) and if so, what to watch out for?

4. As outlined in the introduction, the method may find application in a wide range of fields and settings (e.g. in profiling, on moving platforms, ...). Among them are scenarios, where there is a difference in scale across the time series, both in expected dynamics (e.g., deep ocean with less dynamics vs. surface layer with more dynamics) or in measurement resolution (e.g., a lower and a higher resolution $\Delta t_j$ part).

   Please add some comment on how to deal with such irregularly spaced scenarios: Would you recommend to split such a time series and use a specific model $\Delta t_i$ for different parts for better resolution, or rather keep it as one (with uniform scale $\Delta t_i$) for better L-curve analysis? Or instead of an evenly sampled grid $t_i$ (l. 71) stitch a few pieces with different $\Delta t_i$ sample spacing together?

I understand that generic recommendations are hard to do, so if you want, imagine a profiling scenario in the open ocean where response time is on the order of 100 s and measurement resolution varies from 500 s from 2000 m to 1000 m depth (ca. 20 samples), to 100 s from 1000 m to 300 m depth (ca. 80 samples), and finally to 20 s from 300 m depth to the surface (ca. 150 samples).

5. Side question: Can a $\Delta t_i$ smaller than the max. $\Delta t_j$ be selected by the regularization? From your example in Figure 3d it seems like it.

6. With your approach in general, is there a difference between full series or subseries application (given same parameters and neglecting edge effects)? I.e., in case of very long time series (and thus a large matrix $\mathbf{G}$), is there a trade-off of splitting the time series to avoid memory problems?

7. Code and data availability: Providing the code of this manuscript will be of tremendous help to anyone trying to apply this excellent work. For code repository, I would however discourage a "static" manuscript supplement, which does not allow for bug fixes or feature updates, and instead use a code repository where this is possible (e.g., github/zenodo or similar).

**Minor comments and typos**

- l.69: Please add a reference (e.g., to a textbook?)

- l.106: "The constant $\gamma \gg \sigma_j^{-1}$ is a numerically large weighting constant" Like 10x numerically large or like 1000x numerically large? (Does it matter?) Please add some guidance.

- Eq. 8: $k_i + (\Delta t)^{-1}$  when  $i = 1$ and $j = i$   Typo?

- Eq. 15: This is not normalized by $N$ (as $N$ changes with different $\Delta t_i$), correct?

- l.175: remove "is"

- l.211: Worth adding an equation for this (like Eq. 15)?

- l.242(ff.): Please check the authors guidelines on how to format numbers in exponential notation (I'd have expected $5.747 \cdot 10^{-4}$ s$^{-1}$ or similar).

- l.248: Correct equation referenced?

---

## Author Comment (AC1)

**Author reply to "Comment on gi-2021-28" on GI-2021-28 by Anonymous Referee 2 submitted on on 3 Feb 2022**

Knut Ola Dølven[1], Juha Vierinen[2], Roberto Grilli[3], Jack Triest[4], and Bénédicte Ferré[1]

[1]Centre for Arctic Gas Hydrate, Environment, and Climate,UiT The Arctic University of Norway, Tromsø, Norway
[2]Institute for Physics and Technology, UiT The Arctic University of Norway, Tromsø, Norway
[3]CNRS, University of Grenoble Alpes, IRD, Grenoble INP, 38000 Grenoble, France
[4]4H-JENA engineering GmbH Wischhofstrasse 1-3, 24148 Kiel, Germany

**Correspondence:** Knut Ola Dølven (knut.o.dolven@uit.no)

We thank the referee for a thoughtful review, which has helped us improve the quality of the paper. Text from "Comment on gi-2021-28" on GI-2021-28 by Anonymous Referee 2 are in grey italic font and our responses are in black normal font. Text we added to the manuscript are in emboldened font in quotation marks.

**Reply to the views proposed by reviewer 2**

5 *1. There are too many curves in Figure 3. it is difficult to see clearly;*

The curves represent the amplified noise and uncertainty in the reconstructed signal and it is certainly difficult to see what this signal is supposed to show due to the noise. The very noisy graph in Figure 3a looks noisy with intent, since illustrates a case where there is too much noise amplification, and it is important that this figure remains.

*2. The experimental results of the paper show that the real response signal can be extracted from the measurement signal*
10 *of slow response sensor to eliminate the influence of transmembrane effect, which is in good agreement with the measurement results of DTB sensor. However, the experimental results of the algorithm are introduced in the summary. It is not understood that the correlation R has increased from 0.18 to 0.91. Because the slow response curve is very different from the fast response curve, the correlation between the two must be very low. The correlation between the fast response signal extracted from the slow response signal and the fast response signal measured directly must be very high. It doesn't feel that it can be said to be*
15 *"improved", nor can it reflect the advantage of this algorithm to obtain the fast response signal;*

The reviewer brings up a good point about the applicability of the Pearson correlation coefficient (or coefficient of determination - the Pearson correlation coefficient squared, i.e. the $R^2$). The metric of comparison we used ($R^2$) has, as any such metric, drawbacks (see e.g. Barrett, 1974). One of the drawbacks are its limitations in inferring causality. We have made changes to the

manuscript to clarify that the correlation coefficient only indicates that the two time series are more similar (see for instance l.284). The key requirement for using the $R^2$ in such an application is that the model is validated beforehand. We validated that the technique work (in principle) via simulations and a controlled laboratory settings. These tests are presented prior to the field application which makes the $R^2$ applicable in our case. The simulation and controlled laboratory test are documented in sections 2 and 3. We also used the Mean Absolute Error to supplement $R^2$ as a metric for comparison.

*3. This paper mainly analyzes the influence of time step on the stability of the algorithm. Are there other factors?*

We will address this view under view 4., since we believe these two points are related in both scope and content.

*4. Based on the relevant knowledge of slow response and fast response sensors, is it a good way to directly measure fast response signals? Or is it better to extract from slow response?*

These are indeed very relevant points to address (viewpoint 3 and 4). There are several factors that affect how much temporal resolution can be recovered. The two factors are the decay time, and the measurement noise variance. The slower the sensor, the more difficult it is to recover high temporal resolution fluctuations. The measurement noise also affects how well the deconvolution can be done. The purpose of the L-curve procedure is to determine how good of a time resolution can be obtained. This procedure finds a balance between how noisy the reconstruction is, and how well the reconstruction agrees with the measurements. We have made changes to the manuscript to hopefully explain these factors more clearly for instance in l.131 where we added in: **"The quality of the solution relies on an appropriate choice of regularization parameter $\Delta t$ and noise/uncertainty in the measurements, but also on the ratio between the RT of the sensor and variance in the property of interest."**

**References**

Barrett, J. P.: The Coefficient of Determination—Some Limitations, The American Statistician, 28, 19–20, https://doi.org/10.1080/00031305.1974.10479056, 1974.

---

## Author Comment (AC3)

**Author reply to "Comment on gi-2021-28" on GI-2021-28 by Anonymous Referee 1 submitted on on 13 Dec 2021**

Knut Ola Dølven[1], Juha Vierinen[2], Roberto Grilli[3], Jack Triest[4], and Bénédicte Ferré[1]

[1]Centre for Arctic Gas Hydrate, Environment, and Climate,UiT The Arctic University of Norway, Tromsø, Norway
[2]Institute for Physics and Technology, UiT The Arctic University of Norway, Tromsø, Norway
[3]CNRS, University of Grenoble Alpes, IRD, Grenoble INP, 38000 Grenoble, France
[4]4H-JENA engineering GmbH Wischhofstrasse 1-3, 24148 Kiel, Germany

**Correspondence:** Knut Ola Dølven (knut.o.dolven@uit.no)

The authors would like to express their thanks for this very thorough, positive and constructive review. Here follows our response to the review report. Text from "Comment on gi-2021-28" on GI-2021-28 by Anonymous Referee 1 are in grey italic font and our responses are in black normal font. Text added to the manuscript is written in emboldened font and are in quotation marks.

5 **Comments**

*1. In your simulated data and toy model, you use a measurement error proportional to um plus a constant noise floor term. After reading the full paper, I see where this comes from (your particular application/example), but do not think that it serves your objectives here as outlined in section 2.1 (l. 134ff.). While the lagged inflation/deflation of the estimate's ^ua uncertainty is well explained and understandable in the field example (Figure 6d and corresponding text), it raises question marks and*

10 *creates wrong conclusions here, early on in your derivation, e.g., "So it seems that the method accumulates/amplifies noise terms with time, as the ^ua uncertainty envelope grows and grows with time (Figure 3a and c). What would happen if there were a longer 0-period of say time 50 instead of time 5, would the error envelope be huge already at the step change? Can the method only be applied to short pieces of a longer time series??" Instead, I'd suggest to use only a constant measurement error term in the toy model for simplicity, to illustrate that your method does not suffer from accumulation of noise and (in*

15 *such a case) can keep a nearly constant $\hat{u}_a$ uncertainty. However, ...*

Section "2.1 Simulation and $\Delta t$ determination" has a threefold aim: A) Validate the method in a numerical experiment B) explain how the method is implemented in practice. C) Describe how regularization optimization is implemented (L-curve). An important property of this method is its ability to weigh the least squares solution to Eq. 6 by the measurement uncertainty of each measurement (uncertain measurements are weighted less and vice versa). In our aim to validate the method in a numerical

20 experiment (A) we therefore think that the simulated data should contain varying uncertainty. However, the point made by the referee is valid considering that this section also aims to explain the practical implementation of the method (e.g. considering Figure 3a). We therefore added a parenthesis in the sentence prior to Eq. 14 such that it now reads: **"We assume that measure-**

ment errors are proportional to $u_m(t)$ (measurement uncertainty increase with increasing concentration) in addition to a constant noise floor term, providing a standard deviation for each measurement given by:" and also rewrote the sentence found in l.178/l.179 (in GI-2021-28) such that it now reads: "The error estimate, given as 95% confidence intervals ($2\sigma$ uncertainty), is obtained by Eq. 11 and the simulated measurement uncertainty (which is proportional to $u_m(t)$, Eq. 14).".

*2. I find it hard to understand and believe that (e.g., with a constant $\sigma_j$ ) the uncertainty of the reconstruction $\hat{u}_a$ does not have/show a dependence on the magnitude of correction (e.g., difference between $\hat{u}_a$ and $\hat{u}_m$). I.e., that the reconstructed signal of a sensor pretty much in equilibrium (in no-gradient regions) should be as certain or uncertain as the reconstructed signal where the sensor experiences a strong gradient (and needs a strong correction). Intuitively, with small corrections required (no-gradient case), sensor noise should not be of paramount importance (just slight amplification, reflected in smaller estimate's uncertainty envelope), whereas with large corrections (strong gradient case), the same level of sensor noise should have a stronger impact on the reconstructed signal (more strongly amplified, so larger estimate's uncertainty envelope), shouldn't it? The figures 2 and 3 tend to support this intuition: "Overshoots" in the reconstructed signal $\hat{u}_a$ tend to be a more prominent issue in strong gradients (here: step change, which violates the smoothness assumption as noted in the Appendix A, agreed. But the same is true if $\hat{u}_a$ has a "smooth" strong gradient within $\Delta t_j$ resolution). Similarly, fit residuals (Figure 2d) show a different character at the step change/strong gradient, where they are more coherent (and more coherently wrong) across multiple $t_j$ (suggesting $\hat{u}_a(t_i)$ uncertainty should be larger), whereas they show only high frequency fluctuations around a zero-mean in the later no-/small gradient (suggesting validity of a smaller $\hat{u}_a(t_i)$ uncertainty). This is probably the most important point that needs to be addressed by the authors (and testifies to the otherwise really excellent work): Should the uncertainty on $\hat{u}_a$ not be larger when there is a large correction compared to a small one?*

As observed in the model fit residuals, there are indeed artifacts in the reconstructed signal where the step change occurs - even with our example of a "good" $\Delta t$ choice ($\Delta t = 1.35$, Figure 2d). Model fit residuals should ideally be roughly normally distributed and without strong systematic patterns, which can indicate errors in the model assumptions or model itself. The model fit residuals can for instance reveal if our assumptions about the growth law equation is correct or tell if the complexity of the model is sufficient to resolve the property of interest. In the step-function case, this is indeed what is happening: The extra wiggle at the step change indicates that in this part of the time-series, the model complexity is insufficient to properly model the signal. Figure AR 1. illustrates this by the growth law frequency response, which essentially is a low-pass filter, and indicated frequency region considered by a model with time-step $\Delta t$: The convolution dampens high frequencies in the input signal, however, events with considerable high frequency components (compared to the rest of the time-series - like the step-change) can still make its mark on the measured data. In our model, we assume that the signal we try to reconstruct only changes linearly with time-steps $\Delta t$ or larger (or that there is no information we consider useful between these points). The result is that the measurements $m'_j$ will deviate from this assumed linearity in $u_m(t'_j)$ (our model assumption) between the

[Figure]

**AR 1.** Drawn conceptual figure showing the frequency response of a growth law equation and domain considered by a model (as outlined in GI-2021-28) with a time-step assumption of $\Delta t$ which is indicated with the green shaded box. When considering the noise and noise penetrating occurrence one can picture that the underlying signal is flat in frequency domain with an event indicated by the yellow circle.

55 modelled points $u_m(t_i)$, thereby resulting in increased model fit residuals here. In Figure 1. of the manuscript, this effect is clearly visible as the discrepancy between the red line (modeled measurements $u_m(t'_j)$) and the red dots (real measurements $m'_j$).

In practice, it will naturally be more difficult to reconstruct a rapidly varying signal, since more correction needs to be applied to resolve this, which in turn puts higher demand on the measuring equipment. However, for the algorithm itself it is
60 not the case that the method is generally less reliable/has a higher uncertainty for rapidly changing signal, e.g. where the most likely solution suggests that the property of interest is constant, a possibility still exists that the property is actually varying within modelled uncertainty and that the instrument is incapable of resolving this due to its accuracy limitations. That being said, artifacts due to poor complexity is more of an issue where variability is high and these regions can be identified in the model fit residual (essentially showing that there is more information to be found there).

65 In practice, there are probably many situations where it is difficult to completely avoid local deviations in the model fit residuals, since time-series often have varying local variance (e.g. parts slowly varying, parts fast varying signal). In the L-curve optimization, $\Delta t$ is chosen as a compromise between model fit residuals and noise amplification for the *whole* time series which can result in insufficient model complexity for high variance regions of the time-series (the degree of this problem of course depends on the capabilities of the sensor, i.e. accuracy and response time and the time-series). Fortunately, such regions
70 are identifiable in the model fit residuals, providing information that can be used in data evaluation to either point out regions with slightly increased uncertainty (if the deviation is relatively small) or to re-evaluate the $\Delta t$ assumption and for instance slightly increase the model complexity. It is also possible to split the time-series and apply different $\Delta t$s to different sections and thereby retain high accuracy for regions with smaller high-frequency components. This of course implies that the end result is a more complicated data set (with different time-steps).

75 To make this aspect clearer we

– expanded and rewrote the end of the paragraph starting from l.177 (in GI-2021-28) to: **"Since we assume that the property of interest changes linearly between our model points, a small irregularity in the model fit residuals remains at the step (Figure 2d) due to the models inability to capture the high frequency components of the step function. In practice, the fit residual irregularity arise due to the discrepancy between $u_m(t'_j)$ and $m_j$ at the step change (the effect is clearly visible in the schematic, Figure 1). Nonetheless, the step-change is well represented within the limits of the resolution provided by the model assumption without very eye-catching fit residuals (Figure 2d)."**

– We also added a paragraph at the end of this section 2.1 (which also touches upon several of the comments addressed here): **"Even though the step function is an unrealistic scenario in a practical application, it is likely that the variability of a measured property can change considerably within a single time-series, for instance in profiling applications. Since the L-curve criterion provides a $\Delta t$ which is a compromise between error amplification and model fit residuals, this can result in a model complexity which is too crude to resolve important high variability sections of the data. Generally, the model fit residuals should be roughly normally distributed, and should not have strong irregularities or systematic patterns. Inspection of model fit residuals can identify sections of the data set with too low complexity (resulting in residual spikes, as shown in the simulation experiment) and considerations can be made such as increased caution in data interpretation. Alternatively, it is also possible to manually increase model complexity for the whole or certain parts of the time-series to reduce the fit residuals in these regions."**

– Additionally, we made some small changes to Figure 1 and a couple of other minor edits throughout (see the tracked changes document).

*3. The authors present examples where there are highly-resolved observations together with a very long time constant. This is arguably a very favourable case for response time corrections, where the high resolution allows for a lot of averaging, thus keeping noise amplification low. The authors should add a comment in their manuscript on more poorly resolved scenarios, where $\tau_{63}$ is closer to $\Delta t_j$ and not orders of magnitude between them ($\tau_{63} \gg \Delta t_j$). Is there a limit (e.g., to the applicability, utility, ...) and if so, what to watch out for?*

It is indeed an advantage when the sensor is able to collect large amounts of data, since this method is based on providing a meaningful separation between noise and useful information in any data point and harness this information to provide a good estimate of the ambient concentration - i.e. more data gives better estimates. There is no fixed limit to the applicability, although it of course becomes meaningless to apply this method when $\tau_{63}$ approaches the characteristic variability in the property of interest. Since this method models error amplification and can be evaluated for general consistency (by model fit residuals) the limit to its applicability will become apparent in each individual case. In other words, performance will degrade with decreasing amount of information and/or measurement accuracy. For instance when $\tau_{63}$ approaches $\Delta t_j$, L-curve optimization will most likely give a solution showing that there is not much more information to retrieve from the measurements. In this

case, if $\tau_{63}$ is also very large and the property of interest is in reality varying considerably, the instrument is probably incapable of providing meaningful information for the property of interest (the RT cannot retrieve information which is not there). Figure AR 2. shows the application on the field data with $10^{-1}$, and $100^{-1}$ of the data ($\Delta t_j$ = 20 s/1300 data points and $\Delta t_j$= 200 s/approximately 130 data points). The figure shows that the amount of variability which is possible to resolve is reduced as the measuring frequency and correspondingly amount of information is reduced (as expected), however, there is no critical point where the performance suddenly drops. If the instrument had higher accuracy or faster response time the resolution of the RT-corrected signal would be of correspondingly higher resolution/quality (and the L-curve criterion would yield a different result). The general advice would be the same for any application of this method: An L-curve resembling an L and roughly normally distributed model fit residuals (and if deviations inspect and/or comment) are good indications, however, limitations on what variability can be resolved is of course dependent on the resolution and quality of the data as well as response time of the instrument. In l.131 we added **"The quality of the solution relies on an appropriate choice of regularization parameter $\Delta t$ and noise/uncertainty in the measurements, but also on the ratio between the RT of the sensor and variance in the property of interest."**. In addition, the additions under Comment 2 are also relevant here.

*4. As outlined in the introduction, the method may find application in a wide range of fields and settings (e.g. in profiling, on moving platforms, ...). Among them are scenarios, where there is a difference in scale across the time series, both in expected dynamics (e.g., deep ocean with less dynamic vs. surface layer with more dynamics) or in measurement resolution (e.g., a lower and a higher resolution $\Delta t_j$ part). Please add some comment on how to deal with such irregularly spaced scenarios: Would you recommend to split such a time series and use a specific model $\Delta t_i$ for different parts for better resolution, or rather keep it as one (with uniform scale $\Delta t_i$) for better L-curve analysis? Or instead of an evenly sampled grid $t_i$ (l. 71) stitch a few pieces with different $\Delta t_i$ sample spacing together?*

*I understand that generic recommendations are hard to do, so if you want, imagine a profiling scenario in the open ocean where response time is on the order of 100 s and measurement resolution varies from 500 s from 2000 m to 1000 m depth (ca. 20 samples), to 100 s from 1000 m to 300 m depth (ca. 80 samples), and finally to 20 s from 300 m depth to the surface ca. 150 samples.*

We have touched on this topic under comment 2 but will further address this in comment 6 since they are related.

*5. Side question: Can a $\Delta t_i$ smaller than the max. $\Delta t_j$ be selected by the regularization? From your example in Figure 3d it seems like it.*

You can choose (manually) to have a $\Delta t_i$ which is smaller than $\Delta t_j$, but this would not be meaningful since the region between $\Delta t_{j+1}$ and $\Delta t_j$ is by our (discrete) definition linear (i.e. there is no information here). This is also the case for

[Figure]

**AR 2.** a) Result of the response time algorithm with automatic $\Delta t$ selection for field data with simulated 20 s and b) 200 s measuring interval (rather than ∼2 seconds) to illustrate hypothetical cases where the instrument records at lower frequency.

our model assumption, i.e. the property of interest can only change linearly between any $\Delta t_{i+1}$ and $\Delta t_i$. A solution where $\Delta t_i < \Delta t_j$ would not be chosen in the L-curve criterion since such a solution is 1. of higher complexity and 2. would have higher error amplification but similar model fit residuals than a solution where $\Delta t_i = \Delta t_j$. Both these aspects would make our model implementation of the L-curve criterion (our measurement and fit residual norm definitions, Eq. 15 and 16 in manuscript) prefer the sparser solution.

*6. With your approach in general, is there a difference between full series or subseries application (given same parameters and neglecting edge effects)? I.e., in case of very long time series (and thus a large matrix G), is there a trade-off of splitting the time series to avoid memory problems?*

It could indeed be beneficial to split and stitch data sets to reduce memory issues and solve high/low variability sections separately. This is entirely possible and there are numerous ways to split and stitch data set to limit edge effects. There are no particular issues when doing this when using the proposed method other than being aware of the increased uncertainty towards the edges of the reconstructed time-series (thus it is advised to have overlapping seams). In general, such an approach becomes relevant when

- The number of data-points in the model ($t_i$) approach $\sim 10^5$. In practice, this depends on the total size of the data set and the difference between $\Delta t_i$ and $\Delta t_j$. In the field data set presented in the manuscript, $m'$ was of length $M = 13700$ while the length of $u$ (modeled measurements) was $N = 498$ for the chosen solution.

- When there are sections in the model fit residuals which show a strong effect of too poor complexity in the model. See also comment 2 here.

- This is also the case for the reviewers example with varying time-steps in the measurements: If there is a need for fast response in parts of the time-series, but not in others, splitting the time-series can be a good option. In general, for a deep ocean profile where there is for instance two different regimes with regards to variability and where it is crucial to resolve the high variability regime it could be advisable to split and stitch the time-series, or manually select a model with enough complexity to resolve the high variability regime and accept unnecessary high uncertainty in the low-variability sections.

The added text referred to under comment 2 should address these concerns in the text. Regarding memory usage, there is also a warning trigger implemented in the code.

*7. Code and data availability: Providing the code of this manuscript will be of tremendous help to anyone trying to apply this excellent work. For code repository, I would however discourage a "static" manuscript supplement, which does not allow for bug fixes or feature updates, and instead use a code repository where this is possible (e.g., github/zenodo or similar).*

We completely agree and there is a github repository for code and data here: https://github.com/KnutOlaD/Deconv_code_data. This is also linked in the manuscript under code and data availability.

**Minor comments and typos**

- *l.69: Please add a reference (e.g., to a textbook?)* Reference added.

- *l.106: "The constant $\gamma \gg \sigma_j^{-1}$ is a numerically large weighting constant" Like 10x numerically large or like 1000x numerically large? (Does it matter?) Please add some guidance.* We expanded this sentence such that it now reads: **"The constant $\gamma \gg \sigma_j^{-1}$ is a numerically large weighting constant (we used $\gamma = 2\Delta t \cdot 10^5 \sigma^{-1}$), which ensures**

**that when solving this linear equation, the solution of the growth-law equation will have more weight than the measurements."**

- *Eq. 8: $k_i + (\Delta t)^{-1}$ when $i = 1$ and $j = i$ Typo?* There was indeed a typo, but I believe the typo was it in the line below where in OS-2021-28 said "$-(\Delta t)^{-1}$ *when $i = 1$ and $j = i + 1$*" which is now corrected to "$\mathbf{(\Delta t)^{-1}}$ **when $\mathbf{i = 1}$ and $\mathbf{j = i + 1}$**", this should now be consistent with Eq. 2.

- *Eq. 15: This is not normalized by N (as N changes with different $\Delta t_i$), correct?* Yes.

- *l.175: remove "is"* We removed the *"is"*.

- *l.211: Worth adding an equation for this (like Eq. 15)* Sure, we added an equation here.

- *l.242(ff.): Please check the authors guidelines on how to format numbers in exponential notation (I'd have expected $5.747 \cdot 10^{-4} s^{-1}$ or similar).* We changed this accordingly.

- *l.248: Correct equation referenced?* Indeed not. We corrected this and now the correct equation should be referenced.

---

## Editor Decision (ED1)

*An editor's review on the version 3 manuscript submitted to*
*"Geoscientific Instrumentation, Methods and Data System"*

Title: **Response time correction of slow response sensor data by deconvolution of the growth-law equation**

Authors: Knut Ola Dølven, Juha Vierinen, Roberto Grilli, Jack Triest, and Bénédicte Ferré

Reviewed by Takehiko Satoh (the handling editor)

Thank you very much for your intensive effort in addressing all the issues arisen by two anonymous reviewers through the peer-review period. I'm grateful to find out that this version 3 manuscript is indeed approaching the state of publication. In this report, as the handling editor, I'm going to make several minor comments plus one major comment. They are mostly related to the ways handling uncertainties of the measurements and/or the restored results. No more reviewers will be needed other than me (the handling editor) in subsequent refinement of the manuscript. In the below, indicated line numbers are of the version 3 manuscript.

L. 107: in the in-line equation of $\gamma$, one representative $\sigma$ is used but how it is obtained from $\sigma_j$ is not explained (although I would guess the largest of $\sigma_j$).

L. 145: Eq. (13) is identical with Eq. (3). Avoid repeating the same equation to appear in different places with different numbers (this is not a lengthy paper) but just mention here "The simulated measurements $m_j$ are obtained according to Eq. (3)" or alike.

L. 182-183: "The error estimate is obtained from Eq. 11" needs to be explained more explicitly. My understanding is that the diagonal elements of the upper left quadrant ($N$ x $N$) of $\Sigma_{MAP}$ (the covariance matrix) are used, right? Explain in the text.

L. 184: add "except an offset $\sigma_0$" after "proportional to $u_m(t)$"

L. 232: not very clear why re-definition of $\sigma_j$ is needed instead of using that in Eq. (14). Please clarify.

L. 232: I believe this (equation about $\mu$) reduces to

$$\mu = \frac{1}{M-1}(m_M - m_1)$$

L. 246: "with process 2 inverted" may not be appropriate as processes 2 and 3 are no longer discernible (or, in other word, the second derivative of the curve never becomes null) in the step decrease phases after ~2 x $10^4$ s (Fig. 4c). Better rewording this.

L. 313-326 and Figure 6: I have one *major* concern here about the uncertainties estimated for $u_a(t)$ displayed in Fig. 6d. As the authors noted that "lower for increasing concentrations … at ~00:15", the behavior of uncertainties in Fig. 6d (almost proportional to instantaneous values of "already

convolved" measurements, $m_j$, in Fig. 6a) is far from one would expect. Since the response time of the EB sensor is longer than 2000 s (as in L. 265-266, for the temperature range of field experiment), reconstruction of $u_a(t)$ at a time actually refers all subsequent measurements in next thousands of seconds (wouldn't this appear in the covariance matrix?). How come UPs and DOWNs of $u_a(t)$ in last 10 or so minutes (after ~07:15) be reconstructed this good as in Fig. 6a? This also implies that uncertainties of the reconstructed signals should behave like weighted integrals of the measurement uncertainties in the same length of time.

In addition, the RT-corrected EB data deviate from the DRB sensor measurements by the order of tens of ppm at many occasions while the estimated errors of the former are mostly smaller than 3 ppm except 00:15 – 00:45 after the largest peaks of methane concentrations. To my eyes, attributing ALL of these discrepancies to the different characteristics of the EB sensor and the DRB sensor would not be appropriate but (at least) some should be to suspected under-estimate of the uncertainties of $u_a(t)$.

Although I'm not 100 % sure how this could happen, one possible area may be the authors' choice of numerically large weighting constant ($\gamma$) for the growth-law equation, the upper part in Eq. (6). It seems that a factor $10^5$ in $\gamma$ has rather arbitrarily been chosen. What if this factor is much smaller, say $10^3$, $10^2$ or even unity? I'm very curious whether or not smaller factors could yield "more understandable" behavior of uncertainties for $u_a(t)$ which may make me (and readers) feel probably more comfortable.

---

## Author Response (AR2)

**Author's response to the editor remarks on version 3 of GI-2021-28**

Knut Ola Dølven[1], Juha Vierinen[2], Roberto Grilli[3], Jack Triest[4], and Bénédicte Ferré[1]

[1]Centre for Arctic Gas Hydrate, Environment, and Climate,UiT The Arctic University of Norway, Tromsø, Norway
[2]Institute for Physics and Technology, UiT The Arctic University of Norway, Tromsø, Norway
[3]CNRS, University of Grenoble Alpes, IRD, Grenoble INP, 38000 Grenoble, France
[4]4H-JENA engineering GmbH Wischhofstrasse 1-3, 24148 Kiel, Germany

**Correspondence:** Knut Ola Dølven (knut.o.dolven@uit.no)

We would again like to express our thanks for the impeccable handling and review of our manuscript by the editor and his team and complement the comments which have really helped improving this manuscript. We hope we have interpreted the comments correctly and addressed them appropriately herein and in the revised version of the manuscript.

Text from "An editor's review" of version 3 of GI-2021-28 are in grey italic font and our responses are in black normal font. Text added to the manuscript is written in emboldened font and are in quotation marks. Line numbers referred to in this document might be slightly off due to changes in document length in the review process.

**1 Minor comments**

- *L. 107: in the in-line equation of $\gamma$, one representative $\sigma$ is used but how it is obtained from $\sigma_j$ is not explained (although I would guess the largest of $\sigma_j$).* To ensure that $\gamma$ is large, we have used the smallest $\sigma_j$. The reason for this is that artifacts more likely occur when decreasing $\gamma$ compared to increasing it by a similar amount. See also reply to the major comment below. We changed the text inside the paranthesis to **"we used $\gamma = 2\Delta t \cdot 10^5 \sigma_s^{-1}$, where $\sigma_s$ is the smallest $\sigma$ in $m'$"**

- *L. 145: Eq. (13) is identical with Eq. (3). Avoid repeating the same equation to appear in different places with different numbers (this is not a lengthy paper) but just mention here "The simulated measurements $m_j$ are obtained according to Eq. (3)" or alike.* We removed the equation and changed the sentence introducing the equation to: **"A simulated set of measurements $m_j$ was then obtained by sampling from $u_m(t)$ and adding Gaussian noise $\xi_j \sim \mathcal{N}(0, \sigma_j^2)$ (see Eq. 3"**.

- *L. 182-183: "The error estimate is obtained from Eq. 11" needs to be explained more explicitly. My understanding is that the diagonal elements of the upper left quadrant (N x N) of $\Sigma_{MAP}$ (the covariance matrix) are used, right? Explain in the text.* That's right. We have added an appendix showing the $\Sigma_{MAP}$ of from the field experiment - see response to major comment below. We also changed the referenced sentence to: **"The error estimate, given as a 95% confidence interval ($2\sigma$ uncertainty), is obtained from the the diagonal terms of the upper left quadrant of the covariance matrix (the quadrant concerning $[a, a]$, see Eq. 11) using the simulated measurement uncertainties as input variance (which**

**are proportional to $u_m(t)$ except an offset $\sigma_0$, Eq. 13)."** We also expanded on the sentence following Eq. 11 ($\sim$L. 130) such that it now reads: **"This essentially achieves a mapping of data errors into model errors, including the prior assumption of smoothness."**

- *L. 184 add "except an offset $\sigma_0$" after "proportional to $u_m(t)$"* Yes, added this.

- *L. 232 not very clear why re-definition of $s_j$ is needed instead of using that in Eq. (14). Please clarify.* Added and rewrote the middle part of this section to make the reasoning behind this clearer. It is unnecessary to use the wide error margins provided in the sensor data sheet (which takes into account a wide range of potential measurement errors) in a controlled environment with a freshly calibrated sensor. Furthermore, it illustrates the possibility of calculating sensor noise directly from the sensor signal. The passage now reads (included the sentence prior to and after the changes for context): **"The RT of the sensor was determined to 23 minutes at 22$^o$C (k=0.000725s$^{-1}$) prior to the experiment following standard calibration procedure of the sensor, and parameters affecting membrane permeation was controlled (i.e. water flow rate over membrane surface and water temperature). Running a calibrated sensor in a controlled environment assuming random errors, we chose to use signal noise as basis for measurement uncertainty (rather than as stated in the instrument data sheet). This was estimated using the standard deviation of the single point finite differences in the measurement data, calculated as**

$$\sigma_{j=1,2,3,\dots M} = \sqrt{\tfrac{1}{M-1}\sum_{j=1}^{M-1}\left(m_{j+1}-m_j-\mu\right)^2}, \quad \text{where} \quad \mu = \tfrac{m_{M-1}-m_1}{M-1} \tag{1}$$

**The first 2 hours had lower noise due to the reduced sampling rate and hence longer internal averaging period, but noise was otherwise constant and unrelated to the measured concentration. We therefore used two separate $\sigma$s (measurement uncertainty); one for the first 2 hours and one for the latter part of the measurement period. $\Delta t$ was determined to 179 s using the automatic $\Delta t$ selection based on the L-curve criterion (see Appendix)."**

- *L. 232 I believe this (equation about μ) reduces to $\mu = \frac{1}{M-1}(m_M - m_1)$* It indeed does - we simplified accordingly.

- *L. 246 "with process 2 inverted" may not be appropriate as processes 2 and 3 are no longer discernible (or, in other word, the second derivative of the curve never becomes null) in the step decrease phases after $\sim$2 x 104 s (Fig. 4c). Better rewording this.* Agree. We changed the last sentence in this paragraph to: **"The step decreases behave consistent with the step increases, although process 2 and 3 becomes indiscernible since they both act towards decreasing the concentration."**

**2  Major comment**

*L. 313-326 and Figure 6: I have one major concern here about the uncertainties estimated for $u_a(t)$ displayed in Fig. 6d. As the authors noted that "lower for increasing concentrations ... at 00:15", the behavior of uncertainties in Fig. 6d (almost*

*proportional to instantaneous values of "already convolved" measurements, $m_j$, in Fig. 6a) is far from one would expect. Since the response time of the EB sensor is longer than 2000 s (as in L. 265-266, for the temperature range of field experiment), re-*

55 *construction of $u_a(t)$ at a time actually refers all subsequent measurements in next thousands of seconds (wouldn't this appear in the covariance matrix?). How come UPs and DOWNs of $u_a(t)$ in last 10 or so minutes (after ~07:15) be reconstructed this good as in Fig. 6a? This also implies that uncertainties of the reconstructed signals should behave like weighted integrals of the measurement uncertainties in the same length of time. In addition, the RT-corrected EB data deviate from the DRB sensor measurements by the order of tens of ppm at many occasions while the estimated errors of the former are mostly smaller than 3*

60 *ppm except $00:15 - 00:45$ after the largest peaks of methane concentrations. To my eyes, attributing ALL of these discrepancies to the different characteristics of the EB sensor and the DRB sensor would not be appropriate but (at least) some should be to suspected under-estimate of the uncertainties of $u_a(t)$. Although I'm not 100 % sure how this could happen, one possible area may be the authors' choice of numerically large weighting constant ($\gamma$) for the growth-law equation, the upper part in Eq. (6). It seems that a factor $10^5$ in $\gamma$ has rather arbitrarily been chosen. What if this factor is much smaller, say $10^3$, $10^2$ or even*

65 *unity? I'm very curious whether or not smaller factors could yield "more understandable" behavior of uncertainties for $u_a(t)$ which may make me (and readers) feel probably more comfortable.*

Firstly we would like to comment on that we discovered an error in Figure 6d during the work with this comment. The errors shown in this figure did not previously show 2 standard deviations but only 1 standard deviation. This is now fixed (the error bars and errors reported in the text should be correct). This is of course relevant to the comment above and this discussion in

70 general.

We believe there are several concerns here and we will try to respond to these one by one.

1. The errors we model in our method is confined to the errors in measured concentration within the measuring chamber. This accuracy is given as a percentage (with an offset) of the measured concentration within the measuring chamber. We therefore expect that the model errors to follow and have the same smoothened characteristics as the measured concen-

75 tration within the measuring chamber (the untreated EB data). To be a bit clearer on this, we added a sentence in the last paragraph of section 4.1 that reads **"It is worth noting that this uncertainty regards the gas detection occurring within the measurement chamber and not potential errors caused by other factors such as imperfections with the e.g. membrane or pump operation. "**

As the editor suggests, there are indeed non-diagonal elements in the covariance matrix and to a relatively small degree

80 some temporal relationship within model errors. Figure D1 show the covariance matrix from the field experiment ($\Sigma_{MAP}$ as calculated by Eq. 11 in the manuscript). The upper left quadrant of Figure D1a) show the $(\boldsymbol{a}, \boldsymbol{a})$ covariance and the diagonal elements gives the model uncertainty estimates. The fluctuations on each side of the diagonal elements (non-diagonal elements, see Figure D1b and c) show that there is auto-correlation in the error estimates of $\boldsymbol{a}$. This is expected since the elements in $\boldsymbol{a}$ are constructed from a set of linear combinations of the elements of $\boldsymbol{m}'$ (and $\boldsymbol{G}$, see e.g. chapter

85 2 of Aster et al., (2018) ). In our case, the auto-correlation is quite narrowly confined in time, indicating a relatively short impulse response on our estimate. To explain this (despite a RT of $\tau_{63} >$2000 s), consider that even though the response

time is slow, the effect of a change in the property represented by $a$ is still in principle recorded at the immediate subsequent measurement time-step by producing a change in the shape of $m'$ at that point in time. Knowing the forward operator (the growth-law) makes it possible to interpret this accordingly. Practically speaking, the auto-correlation in the covariance matrix can be viewed as occurring due to the uncertainty in the exact point where the model interprets changes in $m'$ to be a factual change in the property represented by $a$, rather than just caused by a random error. The large increase in estimated uncertainty at each side of the time-series due to lack of measurements (on each side of the time-series) also illustrates the length of the uncertainty estimate impulse response (being relatively short compared to the total time-series).

We have extended and modified the sentence describing how we calculated the uncertainty of the RT-corrected signal in the field experiment (L.314) to read: **"Using the framework of inverse theory allows us to model error behavior by calculating the covariance matrix $\Sigma_{MAP}$ (see Eq. 11 and Appendix D),"** We also added an appendix (Appendix D) on the covariance matrix from the field experiment showing Figure 1 and a short, slightly modified version of the text above: **"Figure 1 show the covariance matrix from the field experiment ($\Sigma_{MAP}$ as calculated by Eq. 11). The upper left quadrant of Figure 1 concerns the $[a,a]$ covariances and the diagonal elements are the model uncertainty estimates. The other quadrants concerns the $[a,u]$, $[u,a]$, and $[u,u]$ covariances. Note that the color-scale in Figure 1 are set to be small to increase the contrast of the plot. The fluctuations on each side of the diagonal elements (non-diagonal elements) in row 100 (see Figure 1b and c) show that there is auto-correlation in the error estimates of $a$. This is expected since the elements in $a$ are constructed from a set of linear combinations of the elements of $m'$ and also, being well confined in time, of little concern (see e.g. Aster et al., (2018) )."**

2. We are very glad the editor pointed this out, since there was indeed a factor 2 error in Figure 6d. This is now fixed. Other than that, the difference between the RT-corrected EB data and DRB sensor exceeds the error margins 8% of the time (this is pointed out in the manuscript) and this should be correct. The reason for this can be many, e.g. imperfections in one or both of the sensors or associated components (e.g. pumps). We comment on this in L. 315 in the manuscript, but we believe our addition in the last paragraph of section 4.1 presented above (in 1.) should help here as well.

3. The choice of weighing constant $\gamma$ can be chosen in this manner because the estimates are very insensitive to the choice of $\gamma$. This is illustrated by Figure 2 which show the reconstruction and error estimate of RT-corrected field experiment data (essentially similar to Figure 6 in the manuscript) using a wide range of $\gamma$ values. Even though the result is very insensitive to $\gamma$, this weighing is crucial to safeguard against violation of our theory assumption the growth law equation (which would essentially render the result worthless). Without the $\gamma$ constant, the least squares solution might end up prioritize a perfect mapping of $u_m$ to $m'$ rather than fully adhere to the growth-law. In practice, this constant cannot be too large, because the least squares solver will not work if the constant is too large. It also must not be too small, because we then allow large errors in the growth law equation solution. However, as long as $\gamma$ is large (relatively speaking) and to some degree balanced by the measurement uncertainty (in our case we define $\gamma = 2\Delta t 10^5 \sigma^{-1}$, i.e. large uncertainty in measurements allows larger residuals in the rows with $\gamma$ in $G$, see Eq. 6 in the manuscript) and the least squares solver

[Figure]

**AR 1.** a) Covariance matrix from response time correction of field experiment data. The $[a, a]$ covariance is shown in upper left quadrant and the lower left and upper right quadrants with very faint but visible diagonals show the $[a, u]$ and $[u, a]$ covariance matrices b) Row 100 of the covariance matrix and c) zoomed in excerpt of row 100 of the covariance matrix (approximate region indicated by the square in c).)

works, the matrix equation should be appropriately weighted. We added a sentence in ∼L. 110 (where $\gamma$ is discussed) that reads: **"Estimates are insensitive to the exact choice of $\gamma$ as long as it is large enough to disallow large errors in the growth law equation solution (which is why it is balanced by $\sigma$)."**

[Figure]

**AR 2.** Concentration and error estimates done with different $\gamma$ values. The dashed/different colored lines (blue, orange, green, and pink) are overlapping, creating a mixed colored graph.

**References**

125   Aster, R. C., Borchers, B., and Thurber, C. H.: Parameter estimation and inverse problems, Elsevier, third edn., 2019.